# TCG: Taming CFG for Flow Matching Models via Moment Matching and Adaptive Clipping

## Abstract

Classifier-free guidance (CFG) is a fundamental technique for flow-based models, significantly enhancing visual quality and prompt adherence. However, the guidance scale is typically tuned empirically due to instability at higher values, which often induces visual artifacts and mode collapse. This paper investigates the underlying mechanisms driving this instability and proposes an effective solution. Our analysis reveals that high CFG scales induce a detrimental distribution shift in the velocity prediction, damaging the generation fidelity. To address this, we introduce **TCG**, a novel plug-and-play method comprising two key components: (1) **Moment Matching (MM)**, which stabilizes the velocity distribution by aligning its first two moments (mean and variance), thereby preventing mode collapse; and (2) **Adaptive Clipping (AdapC)**, which dynamically constrains the guidance update term from both temporal and spatial perspectives to ensure smooth and stable sampling. As a result, our method enables robust and high-quality generation across a wide range of guidance scales. Extensive experiments on diverse text-to-image and text-to-video benchmarks validate that our method outperforms both standard CFG and its state-of-the-art variants.

## 1 Introduction

Flow matching models Lipman et al. (2022); Esser et al. (2024); Labs et al. (2025) have emerged as the leading paradigm in generative modeling, setting new standards in image and video synthesis Gao et al. (2025); Wu et al. (2025); Zhang et al. (2025). Their success stems not only from architectural innovations but also from effective guidance methods that steer generation toward user intent. Among these, classifier-free guidance (CFG) Ho & Salimans (2022) is widely used for its effectiveness in improving visual fidelity and prompt alignment.

CFG amplifies the influence of conditioning signals (*i.e.*, text prompts) during iterative denoising through a single hyperparameter: the guidance scale $w$. Intuitively, higher values of $w$ should yield stronger semantic alignment and improved quality. In practice, however, increasing $w$ leads to diminishing returns and usually triggers severe instabilities such as visual artifacts and mode collapse Saharia et al. (2022); Kynkäänniemi et al. (2024); Sadat et al. (2025). These issues limit the upper bound and robustness of diffusion models, especially for strong prompt adherence.

This work investigates the underlying causes of high-CFG instability in flow-based models. We reveal that large guidance scales lead to uncontrolled growth in the CFG update term, inducing a significant **distribution shift in the predicted velocity**. This shift pushes the velocity prediction far outside its expected stable distribution, resulting in degraded outputs. As shown in Figure 1, when $w = 15$, CFG produces overly uniform and stylistically biased images, indicating mode collapse.

To mitigate this issue, we propose **TCG**, a training-free guidance module designed to stabilize the sampling process. TCG comprises two core components. **(1) Moment Matching (MM)**: A moment recalibration scheme applied directly to the velocity prediction. By zero-centering and variance-aligning the guidance term, MM ensures that the updated velocity remains within the expected data manifold, effectively eliminating mode collapse at high guidance scales. **(2) Adaptive Clipping (AdapC)**: A dual-level clipping mechanism that regulates the magnitude of the guidance signal. Temporal clipping enforces monotonic decay of the update norm over denoising timesteps, while spatial clipping suppresses local outliers in feature space, collectively ensuring stable generation. As shown in Figure 1, TCG improves the quality on both moderate and high guidance scales.

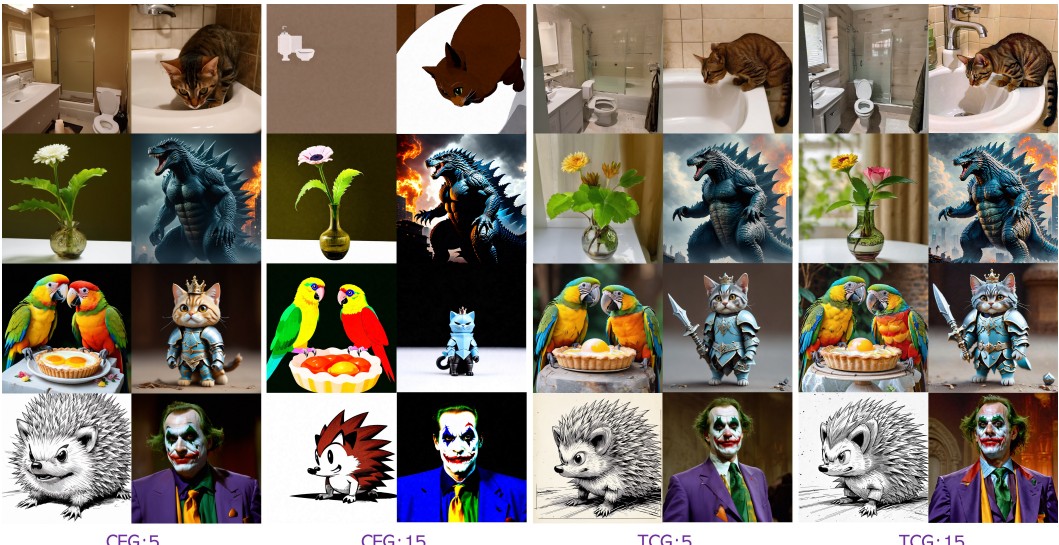

CFG:5      CFG:15      TCG:5      TCG:15

Figure 1: Comparisons between CFG and TCG at different guidance scales. At high scales ($w = 15$), CFG tends to generate overly simplified and stylized (*e.g.*, anime-like) images, indicative of mode collapse. In contrast, TCG produces richer details while preserving output diversity. Results are generated using SD3.5 Esser et al. (2024) with the same random seed.

Consequently, our method unlocks the potential of CFG across a wider range of scales, allowing for better prompt alignment and generation quality. Our contributions can be summarized as follows:

- We provide an analysis that identifies the detrimental distribution shift in the predicted velocity as the key cause of instability at high CFG scales.

- We propose an effective plug-and-play module (**TCG**) for flow-based models, combining a Moment Matching (**MM**) scheme and an Adaptive Clipping (**AdapC**) mechanism, to stabilize the guidance process. This enables robust performance across a wide range of CFG scales and improves the performance upper bound.

- We apply TCG to different SOTA models Esser et al. (2024); Zhuo et al. (2024); Labs (2024); Labs et al. (2025); Wan et al. (2025). Experimental results on diverse image and video generation benchmarks demonstrate that our approach outperforms standard CFG and recent state-of-the-art variants.

## 2 RELATED WORK

### 2.1 FLOW MATCHING DIFFUSION MODELS

Diffusion models have set a new benchmark for high-fidelity image and video synthesis. Early advances Song & Ermon (2019); Song et al. (2020b); Sohl-Dickstein et al. (2015); Nichol et al. (2021); Blattmann et al. (2023) are predominantly SDE-based, with methods such as DDPM Ho et al. (2020), DDIM Song et al. (2020a), EDM Karras et al. (2022; 2024), Stable Diffusion Rombach et al. (2022); Podell et al. (2023); Lin et al. (2024), and DiT Peebles & Xie (2023) modeling stochastic diffusion dynamics via SDEs. More recently, flow-based approaches grounded in flow matching Lipman et al. (2022) have emerged as the mainstream: they formulate generation as a deterministic ODE by learning a time-dependent velocity field that transports samples from a simple prior to the data distribution, leading to more stable training and improved interpretability. Building on this perspective, a series of text-to-image models, including Rectified Flow Liu et al. (2022), SD3/SD3.5 Esser et al. (2024), Lumina-Next Zhuo et al. (2024), and Flux Labs (2024); Labs et al. (2025), as well as text-to-video models Guo et al. (2023); Ma et al. (2025); Team (2024); HaCohen et al. (2024) such as HunyuanVideo Kong et al. (2024) and Wan2.1/2.2 Wan et al. (2025) employ velocity-based training and sampling. Accordingly, our study centers on flow-based models as the primary vehicle for analysis and method design.

## 2.2 Classifier-free Guidance (CFG) for Diffusion Models

Aligning text prompts with image and video generations remains a central yet challenging problem. Early methods used classifier guidance Dhariwal & Nichol (2021), injecting gradients from an external classifier. This approach induces training and compatibility overhead. Classifier-free guidance (CFG) Ho & Salimans (2022) removes the external classifier by jointly training conditional and unconditional models and blending their predictions at inference via a tunable guidance scale. However, this scale is an empirical hyperparameter whose mis-specification can cause artifacts or under-conditioning. To address these issues, some works Zheng & Lan (2023); Xia et al. (2025); Wang et al. (2024); Yehezkel et al. (2025) introduce adaptive or time-varying schedules to improve the guiding process. Some other works Sadat et al. (2023); Kynkäänniemi et al. (2024) focus on enhancing the diversity of generations. Other approaches like Kynkäänniemi et al. (2024) limit guidance to specific sampling intervals. Further refinements to CFG include APG Sadat et al. (2025), which decomposes the CFG update term into parallel and orthogonal components and removes the parallel component to reduce oversaturation. CFG++ Chung et al. (2025) reformulates text-guidance as an inverse problem with a text-conditioned score matching loss, thereby tackling the off-manifold challenges inherent in traditional CFG. More recently, to improve flow-based models, CFG-Zero Fan et al. (2025) optimizes the scale by velocity projections and proposes zero-initialization for the first few steps. In summary, the evolution of text-guided generation techniques highlights a continuous effort to achieve more precise, efficient, and robust alignment between textual prompts and visual outputs. While progress has been made, most methods still struggle with stability at high guidance scales. Our work complements these efforts by targeting the statistical properties of the velocity field, a perspective unexplored in prior studies.

## 3 Method

### 3.1 Moment Matching (MM) for Velocity Stabilization

We begin by analyzing classifier-free guidance (CFG) within the velocity prediction framework of flow-based models. Let $x_1$ denote the clean latent of an image or video, and let $x_0 \sim \mathcal{N}(0, I)$ be a standard Gaussian noise. At timestep $t$, the conditional velocity prediction, guided by a text prompt $y$, is given by $v_t(x|y) = x_0 - x_1^c$, where $x_1^c$ can be regarded as the model's clean latent prediction under prompt condition. The unconditional velocity prediction is similarly formulated as $v_t(x) = x_0 - x_1^u$. The velocity $v_t$ is updated using the standard CFG formula:

$$v_t = v_t(x) + w \cdot (v_t(x|y) - v_t(x)) \tag{1}$$

where $w$ is the guidance scale. The core guidance term, $\delta_v = v_t(x|y) - v_t(x)$, can be rewritten as:

$$\delta_v = (x_0 - x_1^c) - (x_0 - x_1^u) = x_1^u - x_1^c \tag{2}$$

Intuitively, since both $x_1^c$ and $x_1^u$ are underlying estimations of the target within the same clean latent space, we expect their difference $\delta_v$ to have a relatively small magnitude. However, a large guidance scale $w$ can significantly amplify this term. Such amplification can induce a detrimental distribution shift in the final velocity prediction $v_t$, compromising its fidelity to the learned data manifold and ultimately leading to visual artifacts and mode collapse. This observation raises a crucial question: **how can we preserve the effective directional guidance of $\delta_v$ while mitigating the adverse statistical shifts it induces at high guidance scales?**

We reveal that the instability at high CFG scales stems from a distributional mismatch of the predicted velocity. Specifically, while the directional information embedded in $\delta_v$ is crucial for guidance, its statistical moments (mean and variance) can become misaligned with the expected distribution of velocities on the data manifold. Therefore, we introduce a Moment Matching (MM) scheme that explicitly adjusts the first two moments (mean and variance) of the guidance term $\delta_v$.

**Zero-Centering the Guidance Term.** We first hypothesize that the mean component of $\delta_v$, $\mu_\delta = \mathbb{E}[\delta_v]$, contributes little to effective guidance while introducing an adverse mean shift in the final velocity $v_t$. To confirm this hypothesis, we perform zero-centering on the update term, setting $\delta_v^{zc} = \delta_v - \mu_\delta$. Thus the velocity is $v_t^{zc} = v_t(x) + w \cdot \delta_v^{zc}$. Our experiments, as illustrated in Figure 2 and Table 9, confirm that using $\delta_v^{zc}$ for guidance not only prevents degradation in generation quality but can also lead to improvements. This suggests that the mean shift $\mu_\delta$ is dispensable and detrimental. We call this process zero-centering.

**Moment Matching.** Despite the improvements from zero-centering, we observe that for large $w$, the variance of the final velocity $\boldsymbol{v}_t$ can still significantly deviate from that of the unconditional velocity $\boldsymbol{v}_t(\boldsymbol{x})$. This remaining variance mismatch can still contribute to instability. To further stabilize the velocity distribution, we propose to additionally align the variance of the guidance term. Our full Moment Matching (MM) approach combines zero-centering with variance alignment. Therefore, we have:

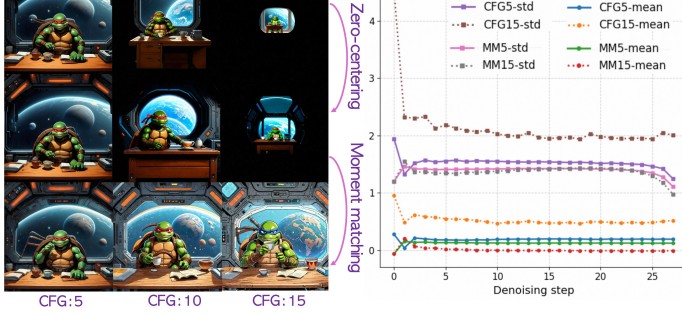

$$\boldsymbol{v}_t^{\text{mm}} = \frac{\boldsymbol{v}_t^{\text{zc}} - \mu}{\sigma_1} \cdot \sigma_2 + \mu \quad (3)$$

Figure 2: Effect of zero-centering and moment matching. Removing the mean leads to better results and full moment matching avoids mode collapse. The right plot shows the velocity prediction statistics across denoising steps. High guidance leads to a distribution shift on the first two moments (mean and variance), while moment matching rectifies this shift. Best viewed in color.

where $\mu = \mathbb{E}[\boldsymbol{v}_t^{\text{zc}}] = \mathbb{E}[\boldsymbol{v}_t(\boldsymbol{x})]$, $\sigma_1 = \text{std}(\boldsymbol{v}_t^{\text{zc}})$, $\sigma_2 = \text{std}(\boldsymbol{v}_t(\boldsymbol{x}))$. This moment matching process aims to preserve the essential guidance of CFG while explicitly controlling the mean and variance of the updated velocity to better align with the learned in-domain distribution. In this way, we stabilize the statistical properties of the guidance, enabling more robust generation at high CFG scales. As shown in Figure 2, MM corrects the biased distribution of original CFG and works well across different guidance scales.

## 3.2 ADAPTIVE CLIPPING

To further mitigate artifacts caused by excessive guidance, we introduce **Adaptive Clipping (AdapC)**, a method designed to dynamically regulate the CFG update term $\delta_v$ by clipping outliers at both temporal and spatial levels.

First, we consider the temporal dynamics of the denoising process. We illustrate the denoising process using CFG in Figure 3. At early stages (high noise levels), both the conditional and unconditional predictions, $\boldsymbol{x}_1^c$ and $\boldsymbol{x}_1^u$, are noisy. As denoising proceeds, the signal-to-noise ratio of the input latent increases, causing the conditional and unconditional predictions to converge. Consequently, the magnitude of the guidance term, $||\delta_v|| = ||\boldsymbol{x}_1^u - \boldsymbol{x}_1^c||$, is expected to monotonically decrease. To enforce this behavior and prevent sudden spikes in guidance, we propose a temporal clipping strategy. At timestep $t$, we clip the magnitude of the current guidance term, $\delta_v^t$, so that it does not exceed the magnitude of the guidance term from the previous denoising step, $\delta_v^{t+1}$, as follows:

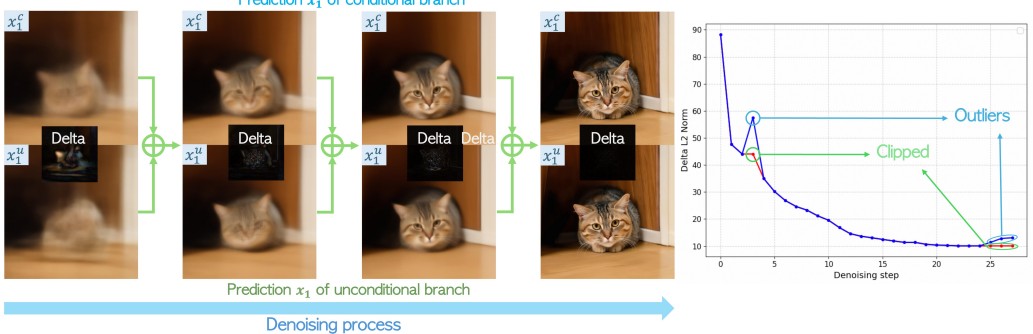

Figure 3: Temporal clipping. During the denoising process using CFG, the delta (the difference between the unconditional and conditional outputs) becomes smaller as the prediction converges. We penalize the outliers, which have a large L2 norm compared to the previous denoising step, to maintain a smoother denoising process. Prompt: *A cat in a house.*

$$\hat{\delta}_v^t = \delta_v^t \cdot \text{clip}\left(\frac{||\delta_v^{t+1}||}{||\delta_v^t||}, 0, 1\right) \tag{4}$$

where $\text{clip}(a, b, c)$ clamps value $a$ between $b$ and $c$. This formula suppresses the norm when $\delta_v^t$ spikes ($||\delta_v^t|| > ||\delta_v^{t+1}||$), and introduces no modification if $||\delta_v^t|| \leq ||\delta_v^{t+1}||$. We call this process **Temporal Clipping (TempC)**. We note that the initial denoising steps are unstable, and thus our clipping strategy skips the first $T_{\text{clip}}$ steps. We set $T_{\text{clip}} = 1$ for models using 28 denoising steps (SD3.5) and $T_{\text{clip}} = 3$ for models using 50 denoising steps (Flux-dev).

However, this temporal clipping may be insufficient to address localized artifacts. The guidance term $\delta_v$ can exhibit high-magnitude values at specific spatial locations (*i.e.*, local outliers), even when its overall norm is reasonable. To address this, we introduce a **Spatial Clipping (SpaC)** mechanism. This method limits the local guidance strength relative to the magnitude of the unconditional velocity prediction at the same location. For an image latent at spatial index $(i, j)$, the clipped guidance term is computed as:

$$\delta_v^{i,j} = \hat{\delta}_v^{i,j} \cdot \text{clip}\left(\frac{||v^{i,j}(x)||}{\gamma w \cdot ||\hat{\delta}_v^{i,j}||}, 0, 1\right) \tag{5}$$

where $w$ is the guidance scale, $\gamma$ is a tunable hyperparameter controlling the clipping threshold, and we set it to 1.5 by default. As shown in

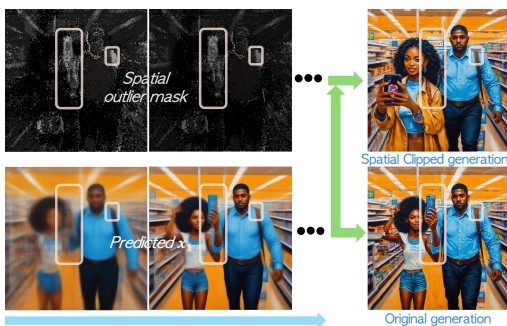

Figure 4: Spatial clipping. Outlier responses often correspond to artifact-prone regions. Clipping based on local regions suppresses these without affecting valid features. Prompt: *A painting depicting a black woman taking a selfie in Wal-Mart while being followed by a man.*

Figure 4, this spatial clipping approach effectively reins in local outliers without suppressing valid guidance in other regions, thereby removing artifacts while enhancing stability.

## 4 EXPERIMENT

Table 1: Quantitative comparisons on HPD v2 benchmark. "G.S." is guidance scale; "P.S." is PickScore; "Aes." is aesthetic; "I.R." is ImageReward; "U.R." is UnifiedReward.

| Model | G.S. | P.S. | Aes. | CLIP | HPS | I.R. | U.R. |
|---|---|---|---|---|---|---|---|
| CFG | | 22.78 | 5.984 | 37.03 | 29.66 | 1.0818 | 3.3988 |
| CFG++ | | 20.93 | 5.814 | 32.75 | 23.98 | -0.0159 | 2.5197 |
| APG | 5.0 | 21.82 | 5.985 | 35.06 | 24.81 | 0.4964 | 2.9532 |
| CFG-Zero | | 22.84 | 6.014 | 36.89 | **30.31** | 1.0876 | 3.4190 |
| **TCG** | | **22.95** | **6.022** | **37.26** | 30.22 | **1.1126** | **3.4230** |
| CFG | | 22.44 | 5.866 | 36.57 | 29.21 | 1.0361 | 3.3662 |
| CFG++ | | 22.26 | 6.020 | 36.37 | 28.23 | 0.8044 | 3.1727 |
| APG | 10.0 | 22.42 | 6.040 | 36.24 | 27.37 | 0.8606 | 3.2494 |
| CFG-Zero | | 22.72 | 5.972 | 37.00 | 30.64 | 1.1558 | 3.4431 |
| **TCG** | | **23.02** | **6.053** | **37.33** | **31.29** | **1.2216** | **3.4958** |
| CFG | | 21.43 | 5.507 | 34.31 | 25.15 | 0.4922 | 2.9521 |
| CFG++ | | 22.60 | 6.051 | 36.98 | 29.76 | 1.0092 | 3.3510 |
| APG | 15.0 | 22.67 | 6.065 | 36.67 | 28.68 | 0.9934 | 3.3685 |
| CFG-Zero | | 22.25 | 5.824 | 36.60 | 29.27 | 1.0363 | 3.2907 |
| **TCG** | | **22.98** | **6.067** | **37.31** | **31.60** | **1.2482** | **3.4812** |

Table 2: Ablation study on MM (Moment Matching) and AdapC (Adaptive Clipping). Both components bring improvements.

| Model | P.S. | Aes. | CLIP |
|---|---|---|---|
| CFG | 22.44 | 5.866 | 36.57 |
| +MM | 22.92 | 6.032 | 37.16 |
| +AdapC | 22.72 | 5.950 | 37.22 |
| +Both | **23.02** | **6.053** | **37.33** |

Table 3: Ablation study on clipping strategies. "SpaC" is spatial clipping and "TempC" is temporal clipping. "AdapC" refers SpaC+TempC.

| Model | P.S. | Aes. | CLIP |
|---|---|---|---|
| CFG | 22.44 | 5.866 | 36.57 |
| +SpaC | 22.71 | 5.935 | 37.21 |
| +TempC | 22.61 | 5.929 | 36.94 |
| +AdapC | **22.72** | **5.950** | **37.22** |

### 4.1 EXPERIMENTAL SETUP

**Implementation Details** We use the official model and implementations for each baseline and base model. Specific implementation details are provided in the appendix.

Table 4: Quantitative comparisons on DPG benchmark Hu et al. (2024). TCG achieves state-of-the-art results on the overall metric and most sub-metrics across different guidance scales.

| Model | G.S. | Global | Entity | Attribute | Relation | Other | Overall |
|---|---|---|---|---|---|---|---|
| CFG Ho & Salimans (2022) | | 84.50 | 90.27 | **88.38** | 93.65 | 82.80 | 84.36 |
| CFG++ Chung et al. (2025) | | 79.79 | 82.89 | 80.39 | 90.35 | 70.40 | 74.70 |
| APG Sadat et al. (2025) | 5.0 | 82.98 | 86.89 | 85.11 | 92.22 | 75.60 | 80.03 |
| CFG-Zero Fan et al. (2025) | | 84.50 | 90.48 | 88.28 | 93.60 | 81.90 | 84.97 |
| **TCG (Ours)** | | **85.11** | **90.64** | 88.37 | **93.71** | **82.90** | **85.16** |
| CFG Ho & Salimans (2022) | | 82.29 | 90.49 | 88.23 | 93.49 | 83.70 | 84.51 |
| CFG++ Chung et al. (2025) | | 84.80 | 87.57 | 85.76 | 91.78 | 78.00 | 81.67 |
| APG Sadat et al. (2025) | 10.0 | **85.33** | 89.22 | 86.97 | 93.38 | 79.10 | 83.09 |
| CFG-Zero Fan et al. (2025) | | 83.43 | 90.92 | 88.62 | 93.82 | 82.30 | 85.29 |
| **TCG (Ours)** | | 84.19 | **91.41** | **88.64** | **94.13** | **85.20** | **85.87** |
| CFG Ho & Salimans (2022) | | 78.34 | 87.44 | 84.52 | 91.59 | 80.10 | 79.93 |
| CFG++ Chung et al. (2025) | | **85.94** | 88.37 | 86.60 | 92.28 | 79.50 | 82.96 |
| APG Sadat et al. (2025) | 15.0 | 85.26 | 90.01 | 87.70 | 93.55 | 80.50 | 84.31 |
| CFG-Zero Fan et al. (2025) | | 81.00 | 89.85 | 87.51 | 93.16 | 82.00 | 83.40 |
| **TCG (Ours)** | | 83.97 | **91.83** | **88.58** | **94.55** | **85.70** | **86.40** |

Table 5: Quantitative comparisons on GenEval benchmark Ghosh et al. (2023).

| Methods | G.S. | Single Object | Two Object | Counting | Colors | Position | Color Attribution | Overall |
|---|---|---|---|---|---|---|---|---|
| CFG | | 99.38 | 83.08 | **65.00** | 81.12 | 23.75 | 47.15 | 66.58 |
| CFG++ | | 76.25 | 39.65 | 25.31 | 45.48 | 8.75 | 13.21 | 34.78 |
| APG | 5.0 | 95.94 | 64.65 | 39.38 | 71.54 | 14.50 | 27.85 | 52.31 |
| CFG-Zero | | 99.69 | 82.58 | 60.94 | **83.51** | 24.00 | **49.39** | 66.68 |
| **TCG** | | **100.00** | **85.10** | 63.44 | 82.18 | **25.50** | 46.14 | **67.09** |
| CFG | | 99.06 | **88.64** | **66.88** | 78.19 | 27.25 | 43.09 | 67.18 |
| CFG++ | | 94.38 | 73.23 | 44.06 | 72.07 | 21.25 | 37.40 | 57.07 |
| APG | 10.0 | 99.38 | 73.99 | 50.94 | 78.72 | 19.25 | 38.21 | 60.08 |
| CFG-Zero | | 99.38 | 85.61 | 64.38 | **82.71** | 25.25 | 46.54 | 67.31 |
| **TCG** | | **100.00** | 87.12 | 62.81 | 80.32 | **27.64** | **52.44** | **68.39** |
| CFG | | 95.31 | 81.06 | 54.69 | 71.01 | 22.25 | 30.69 | 59.17 |
| CFG++ | | 97.81 | 80.56 | 54.37 | 80.59 | 25.00 | 41.46 | 63.30 |
| APG | 15.0 | 99.06 | 81.06 | 50.94 | 81.12 | 22.00 | 44.72 | 63.15 |
| CFG-Zero | | 99.69 | 84.60 | 61.88 | 78.19 | 25.25 | 40.04 | 64.94 |
| **TCG** | | **100.00** | **86.36** | **64.06** | **83.51** | 26.44 | 50.20 | **68.43** |

**Baselines and Base models.** We conduct a comparative analysis not only against the original classifier-free guidance (CFG) but also against three prominent advanced guidance methods: APG Sadat et al. (2025), CFG++ Chung et al. (2025), and CFG-Zero Fan et al. (2025), where CFG-Zero is also designed for flow-based models. Please note that we map the guidance scale for CFG++ to its hyperparameter. For base models in the T2I task, we employ large-scale flow-based models including Stable Diffusion 3 medium (SD3) Esser et al. (2024), SD3.5 medium Esser et al. (2024), Lumina-Next Zhuo et al. (2024), and Flux-dev Labs (2024); Labs et al. (2025). The main experiments and ablations are based on the SD3.5 medium model. Please note that Flux-dev is a CFG-distilled model. We employ different guidance scales to mimic its CFG mechanism, which

Table 6: Comparisons on SD3 base model.

| Model | G.S. | P.S. | Aes. | CLIP | HPS | I.R. | U.R. |
|---|---|---|---|---|---|---|---|
| CFG | 5.0 | 22.64 | 5.956 | **36.41** | 29.64 | 1.0535 | 3.3728 |
| TCG | | **22.73** | **5.998** | 36.34 | **29.84** | **1.0687** | **3.3757** |
| CFG | 10.0 | 22.35 | 5.845 | 36.53 | 29.53 | 1.0717 | 3.3385 |
| TCG | | **22.69** | **5.988** | **36.86** | **30.68** | **1.1462** | **3.3884** |
| CFG | 15.0 | 21.73 | 5.606 | 35.80 | 27.54 | 0.8487 | 3.0708 |
| TCG | | **22.59** | **5.971** | **36.88** | **30.65** | **1.1582** | **3.3464** |

Table 7: Comparisons on Lumina-Next base model.

| Model | G.S. | P.S. | Aes. | CLIP | HPS | I.R. | U.R. |
|---|---|---|---|---|---|---|---|
| CFG | 5.0 | 22.28 | 6.175 | 34.18 | 27.44 | 0.7343 | **2.9659** |
| TCG | | **22.50** | **6.255** | **34.40** | **28.13** | **0.7962** | 2.9634 |
| CFG | 10.0 | 21.65 | 5.972 | 33.41 | 26.08 | 0.5201 | 2.8252 |
| TCG | | **22.52** | **6.231** | **34.88** | **28.74** | **0.8696** | **2.9970** |
| CFG | 15.0 | 21.15 | 5.822 | 32.60 | 24.97 | 0.3301 | 2.6609 |
| TCG | | **22.47** | **6.212** | **35.01** | **28.86** | **0.8772** | **2.9742** |

Table 8: Comparisons on Flux-dev base model. Please note that Flux-dev is a CFG-distilled model. For a fair comparison, we mimic the guidance mechanism.

| Model | G.S. | P.S. | Aes. | CLIP | HPS | I.R. | U.R. |
|-------|------|------|------|------|-----|------|------|
| CFG | 5.0 | 22.87 | 6.009 | 36.88 | 28.90 | 1.1284 | 3.4244 |
| **TCG** | | **23.07** | **6.092** | **37.01** | **29.76** | **1.1827** | **3.4476** |
| CFG | 10.0 | 22.22 | 5.661 | 35.63 | 26.87 | 0.8747 | 3.1173 |
| **TCG** | | **23.03** | **6.071** | **37.29** | **30.25** | **1.2238** | **3.4506** |
| CFG | 15.0 | 21.36 | 5.244 | 33.06 | 23.32 | 0.3743 | 2.6609 |
| **TCG** | | **22.98** | **6.062** | **37.35** | **30.42** | **1.2363** | **3.4340** |

Table 9: Ablation study on zero-centering. "Z-C" refers to zero-centering. It indicates that the mean of CFG update term is not helpful for the generation.

| Model | P.S. | Aes. | CLIP |
|-------|------|------|------|
| CFG | 22.44 | 5.866 | 36.57 |
| +Z-C | 22.58 | 5.903 | 36.91 |
| +MM | **22.92** | **6.032** | **37.16** |

Table 10: Comparisons on Vbench benchmark. We use the recent Wan2.2 models as our base model. Compared to vanilla CFG, TCG improves both frame aesthetics and overall video quality.

| Model | Guidance | Aesthetic Quality | Motion Smoothness | Overall Consistency | Spatial Relationship | Temporal Style | Quality Score | Semantic Score | Total Score |
|-------|----------|-------------------|-------------------|---------------------|----------------------|----------------|---------------|----------------|-------------|
| | CFG 4.0 | 58.69 | **98.69** | 24.81 | 75.38 | 24.81 | 83.02 | 71.19 | 80.65 |
| Wan2.2 5B | CFG 9.0 | 59.09 | 98.22 | 25.36 | **80.67** | 24.82 | 83.36 | **74.74** | 81.64 |
| | **TCG 9.0** | **59.69** | 98.53 | **25.55** | 80.15 | **25.02** | **83.89** | 74.05 | **81.92** |
| | CFG 4.0 | 62.69 | 98.20 | 26.14 | 79.86 | 23.92 | 83.93 | 75.81 | 82.30 |
| Wan2.2 A14B | CFG 9.0 | 62.64 | 97.73 | 26.23 | **80.95** | **24.26** | 83.63 | 76.66 | 82.24 |
| | **TCG 9.0** | **62.82** | **98.23** | **26.24** | 80.54 | 24.13 | **84.07** | **76.76** | **82.61** |

may not be identical to standard CFG results. For T2V tasks, we utilize the latest state-of-the-art Wan2.2 5B and Wan2.2 A14B models Wan et al. (2025).

**Benchmarks.** Our evaluation encompasses both text-to-image (T2I) and text-to-video (T2V) tasks, conducted on different benchmarks. For T2I evaluation, we utilize three prominent benchmarks: HPD v2 Wu et al. (2023), which comprises 3,200 prompts across four styles (animation, concept art, paintings, and photos); GenEval Ghosh et al. (2023), which focuses on object-centric text-to-image generation using compositional prompts to assess the model's understanding of complex relationships; and DPG Hu et al. (2024), which consists of 1K dense prompts, enabling fine-grained assessment of different aspects of prompt adherence. These benchmarks are designed to assess model performance in complex scenes. For T2V evaluation, we adopt the standard prompts and evaluation metrics provided by VBench Huang et al. (2024), which contains around 1K prompts for different dimensions.

**Metrics.** For the standard GenEval, DPG, and VBench benchmarks, we employ their official metrics. For HPD v2, we employ four types of overall human preference metrics: PickScore Kirstain et al. (2023), HPSv2.1 Wu et al. (2023), ImageReward Xu et al. (2023), and UnifiedReward Wang et al. (2025), where UnifiedReward is based on a state-of-the-art VLM model Bai et al. (2025). Furthermore, we use the Aesthetic score Schuhmann (2022) and CLIP score Radford et al. (2021) to measure the aesthetic quality and prompt-following ability, respectively.

## 4.2 QUANTITATIVE EVALUATION

Table 1 presents the quantitative results of our proposed TCG compared to CFG across various methods on HPD v2 benchmark under different guidance scales (G.S.). TCG consistently achieves superior performance. As evidenced by the table, TCG surpasses other methods in terms of Aesthetic Score (Aes.), PickScore (P.S.), HPS, Image Reward (I.R.), and Unified Reward (U.R.) across all guidance scales. Specifically, for a guidance scale (G.S.) of 15.0, TCG significantly improves the CLIP score to 37.31 and the Aesthetic score to 6.067, demonstrating its effectiveness in enhancing both text-image alignment and visual appeal. It also outperforms other methods on human preference metrics: PickScore, HPS, Image Reward (I.R.), and Unified Reward (U.R.). The consistent improvement in Aesthetic Score suggests that TCG produces images with more coherent textures, lighting, and structure, aligning better with human preferences. Moreover, the enhanced CLIP Score confirms that generated images better capture the semantics of the given prompts. The performance on other official benchmarks: GenEval (Table 5) and DPG (Table 4) further highlights the strength of our approach, showing its effectiveness in handling complex generation tasks and refining subop-

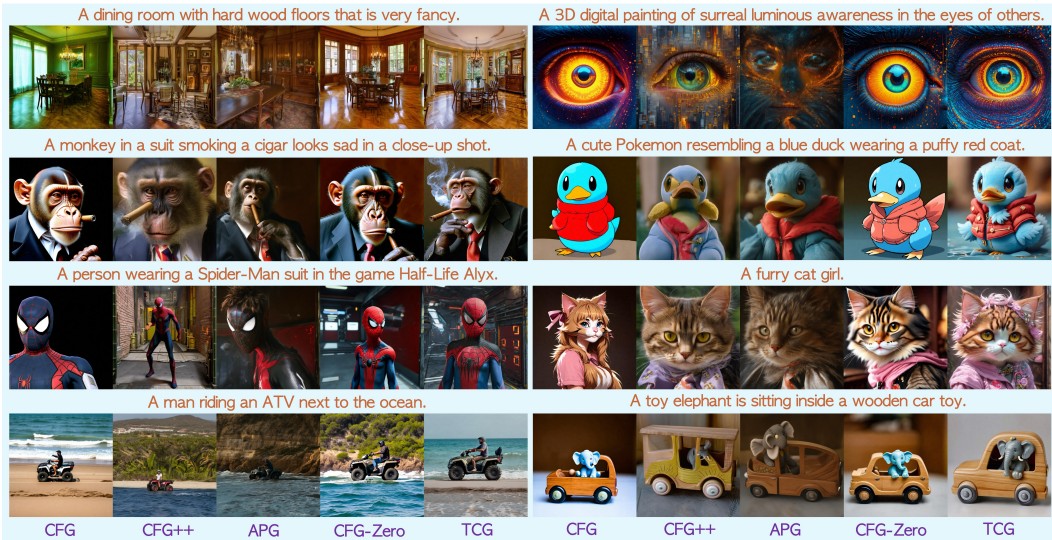

Figure 5: Qualitative comparisons on SD3.5 medium base model at guidance scale 10. TCG obtains more visually appealing and better prompt-aligned results.

timal results produced by CFG. We also verify our method on different base models. As shown in Tables 6, 7, and 8, TCG shows consistent improvements.

To further evaluate the effectiveness and versatility of TCG, we conduct experiments on the text-to-video (T2V) generation task. This evaluation uses state-of-the-art models and standard benchmarks to assess performance. The quantitative results for text-to-video generation are presented in Table 10. Specifically, when applied to the Wan2.2 model (which includes the 5B and A14B versions) Wan et al. (2025), TCG demonstrates marked improvements across several key metrics. It enhances both quality and semantic scores, indicating that TCG boosts video appeal and prompt alignment.

### 4.3 QUALITATIVE EVALUATION

The qualitative comparisons are presented in Figure 5 (and Figure 6 for video), offering a comprehensive visual demonstration of our method's efficacy. TCG consistently produces high-quality images, characterized by rich detail and strong semantic alignment with the given text descriptions. Compared to conventional methods such as CFG and other contemporary approaches, TCG demonstrates significant improvements in both visual fidelity and semantic coherence, findings that are fully consistent with our quantitative experimental results. Furthermore, when applied to video generation, TCG yields more temporally consistent and coherent frames, mitigating artifacts often observed in other methods. This compelling visual evidence not only reinforces the robust performance of TCG but also highlights its potential across various generative tasks. Additional visualizations and extensive qualitative results are provided in the appendix and supplementary material.

### 4.4 ABLATIONS

We provide detailed ablations on different components of TCG. We employ SD3.5 medium base model and a guidance scale of 10 by default for these ablation studies. More ablations are provided in the appendix.

**Impact of Moment Matching (MM) and Adaptive Clipping (AdapC).** Table 2 provides the ablation results for our two main components: Moment Matching (MM) and Adaptive Clipping (AdapC). Each component brings significant improvements. For example, MM improves Aesthetic score from 5.866 to 6.032, and AdapC boosts CLIP score from 36.57 to 37.22. When using both components, our method obtains the best performance across all metrics.

**Impact of clipping strategies.** We show the ablation results for temporal and spatial clipping methods in Table 3, which is denoted by TempC and SpaC, respectively. SpaC makes the velocity more

Figure 6: Qualitative comparisons on Wan2.2 5B base model at guidance scale 9. TCG obtains more consistent and coherent results.

stable and suppresses outliers in a fine-grained manner. It brings significant improvements. TempC focuses on the overall norm of the current denoising step, facilitates better denoising dynamics, which further boosts the results.

**Impact of Zero-Centering.** Table 9 provides the ablation study on removing the mean (zero-centering) from the CFG update term. We observe that zero-centering alone does not harm performance but contributes improvements over original CFG. This verifies our claim from Section 3.1 that the mean component is not helpful to the final results and even detrimental. The full Moment Matching (MM) approach, which includes variance alignment, further enhances performance.

## 5 LIMITATION AND FUTURE WORK

While TCG is simple and consistently improves upon standard CFG across various models and benchmarks, it has several limitations. First, our analysis and design are based on the velocity-prediction framework used in flow-matching models. Although we demonstrate gains on popular flow-based models (*e.g.*, SD3, SD3.5, Lumina-Next, Flux, Wan2.2), TCG has not been exhaustively evaluated on SDE-based models (*e.g.*, DDPM/EDM). Extending TCG to such frameworks may require additional adaptations. Second, our argument for stabilizing velocity distributions is empirical and intuitive; formal theoretical analysis of guidance-induced distribution shifts in deterministic samplers remains an open direction for future work.

## 6 CONCLUSION

In this work, we introduce TCG, a novel training-free guidance method designed to enhance the performance of flow-based models, by improving upon the traditional classifier-free guidance (CFG) mechanism. Our approach addresses the limitations of CFG, which empirical analysis reveals that it can often lead to suboptimal results and artifacts, especially at high guidance scales. TCG incorporates two primary technical innovations: (1) Moment Matching (MM) for distribution calibration of the velocity prediction by using zero-centering and variance alignment, and (2) Adaptive Clipping (AdapC) to stabilize the guidance update term throughout the denoising process at both temporal and spatial perspectives. These components work in concert to guide the model away from potential low-quality predictions, thereby improving overall fidelity. Through comprehensive analysis and extensive experiments, we demonstrate the effectiveness of TCG across both text-to-image (T2I) and text-to-video (T2V) generation tasks. Our evaluations utilize state-of-the-art models such as SD3.5, Lumina-Next, Flux, and Wan2.2, alongside widely recognized benchmarks including HPD v2, GenEval, DPG, and VBench. The results consistently show that TCG outperforms standard CFG, achieving higher aesthetic scores, improved text alignment, and fewer generation artifacts. Furthermore, TCG has been shown to surpass other advanced guidance strategies. The superior performance and robustness of TCG highlight its potential to serve as a versatile and effective method for enhancing the output quality of flow-based models.

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

# Appendix

Due to the space limitation, we provide details omitted in the main text in this appendix, which is organized as follows:

- Section A : Algorithm overview.

- Section B : Ablations on hyperparameters.

- Section C : Detailed implementation for different base models and baselines.

- Section D : Robustness over wide guidance range.

- Section E : More visualization on T2I base models.

- Section F : LLM usage.

For better visualization of video results, please refer to `Visualization_webpage.html` in the supplementary materials.

## A   ALGORITHM OVERVIEW

We provide an overview of TCG in Algorithm 1. It is a plug-and-play module and can be easily implemented for current flow-based diffusion models.

---

**Algorithm 1** The Proposed Guidance Method: TCG.

---

1: **Input:** Velocity prediction $\boldsymbol{v}_t(\boldsymbol{x}), \boldsymbol{v}_t(\boldsymbol{x}|y)$, guidance scale $w$, clipping factor $\gamma$, clipping start step $T_{\text{clip}}$, sampling timesteps $T$.
2: $T_d \leftarrow 0$
3: **for** $t = T$ to $0$ **do**
4:     Compute original guidance: $\delta_v^t = \boldsymbol{v}_t(\boldsymbol{x}|y) - \boldsymbol{v}_t(\boldsymbol{x}), T_d \leftarrow T_d + 1$
5:     # 1. Adaptive Clipping (AdapC)
6:     **if** $T_d > T_{\text{clip}}$ **then**
7:         $\delta_v^{\text{temp\_clip}} \leftarrow \delta_v^t \cdot \text{clip}\left(\frac{||\delta_v^{t+1}||}{||\delta_v^t||}, 0, 1\right)$ # Temporal Clipping (TempC)
8:         $\delta_v^{i,j} \leftarrow \delta_v^{\text{temp\_clip},i,j} \cdot \text{clip}\left(\frac{||v_t^{i,j}||}{\gamma w \cdot ||\delta_v^{\text{temp\_clip},i,j}||}, 0, 1\right)$ # Spatial Clipping (SpaC)
9:     **end if**
10:    # 2. Moment Matching (MM)
11:    $\boldsymbol{v}_t^{\text{zc}} \leftarrow \boldsymbol{v}_t(\boldsymbol{x}) + w \cdot (\delta_{\boldsymbol{v}} - \mu_\delta)$, where $\mu_\delta \leftarrow \mathbb{E}[\delta_{\boldsymbol{v}}]$
12:    $\mu \leftarrow \mathbb{E}[\boldsymbol{v}_t^{\text{zc}}], \sigma_1 \leftarrow \text{std}(\boldsymbol{v}_t^{\text{zc}}), \sigma_2 \leftarrow \text{std}(\boldsymbol{v}_t(\boldsymbol{x}))$.
13:    $\boldsymbol{v}_t^{\text{mm}} \leftarrow \frac{\boldsymbol{v}_t^{\text{zc}} - \mu}{\sigma_1} \cdot \sigma_2 + \mu$
14:    # 3. Solving ODE
15:    $\boldsymbol{x}_t \leftarrow \text{ODEStep}(\boldsymbol{v}_t^{\text{mm}}, \boldsymbol{x}_{t+1})$
16: **end for**
17: **Return** clean latent $\boldsymbol{x}_1$

---

## B   ABLATIONS ON HYPERPARAMETER

In this section, we investigate the impact of hyperparameters in TCG. For $T_{clip}$, it controls the timesteps of applying the proposed adaptive clipping (AdapC) strategy. As shown in Table 11, we can obtain the best aesthetic score when employing AdapC all the time, while it sacrifices the CLIP score because the first several denoising steps are unstable. Thus we set $T_{clip}$ to 1 by default for the model of 28 sampling steps. For $\gamma$, it determines the clipping norm threshold in the spatial clipping part. As shown in Table 12, higher value refers to lower norm, and it will more aggressively clip the norm. We find it is beneficial for semantics, and we set $\gamma = 1.5$ to achieve performance balance.

Table 11: Ablation study on $T_{\text{clip}}$.

| $T_{clip}$ | PickScore | Aesthetic | CLIP | HPS | ImageReward | UnifiedReward |
|---|---|---|---|---|---|---|
| 0 | 23.01 | **6.056** | 37.32 | 31.30 | 1.2200 | 3.4915 |
| 1 | **23.02** | 6.053 | **37.33** | 31.29 | 1.2216 | **3.4958** |
| 2 | 22.99 | 6.043 | 37.27 | 31.30 | 1.2228 | 3.4801 |
| 3 | 22.99 | 6.037 | 37.24 | 31.35 | 1.2264 | 3.4670 |
| 4 | 22.98 | 6.039 | 37.21 | 31.38 | 1.2251 | 3.4726 |
| 5 | 22.98 | 6.038 | 37.20 | **31.40** | **1.2310** | 3.4679 |

Table 12: Ablation study on $\gamma$.

| $\gamma$ | PickScore | Aesthetic | CLIP | HPS | ImageReward | UnifiedReward |
|---|---|---|---|---|---|---|
| 0.5 | 22.94 | 6.038 | 37.21 | 31.42 | 1.2299 | 3.4702 |
| 1.0 | 22.99 | 6.043 | 37.27 | **31.52** | **1.2321** | 3.4824 |
| 1.5 | **23.02** | **6.053** | 37.33 | 31.29 | 1.2216 | **3.4958** |
| 2.0 | **23.02** | 6.048 | 37.40 | 31.06 | 1.1982 | 3.4903 |
| 2.5 | 23.01 | 6.040 | **37.46** | 30.78 | 1.1718 | 3.4772 |
| 3.0 | 22.99 | 6.040 | **37.46** | 30.43 | 1.1450 | 3.4585 |

## C  IMPLEMENTATION DETAILS

### C.1  BASE MODELS

For all base models, we generate $1024 \times 1024$ images, and we follow their official sampling settings, and here we describe related details. For **SD3/SD3.5**, we employ the medium models and use the same sampling setting, namely, denoising steps. For **Lumina-Next**, we employ the official model and use denoising steps 30. For **Flux-dev**, We employ the officially released model. Note that it is a CFG-distilled model, thus we modify its pipeline to mimic standard CFG. Specifically, we set guidance_scale as 1.0 and use the true_cfg_scale in the pipeline, to control its CFG scale. For **Wan2.2 5B** base model, note that it uses a new highly-compressed VAE, we use the recommended resolution of $121 \times 704 \times 1280$ (f, h, w). For **Wan2.2 A14B** base model, it uses a standard video VAE as Wan 2.1. Concerning its high computation cost, we use the recommended resolution of $81 \times 480 \times 832$ (f, h, w). We use the norm dimension 1 for temporal clipping and norm dimensions 3 and 4 for spatial clipping. Namely, we suppress outliers in a more fine-grained manner across subsequent denoising steps. We find that this manner is more stable for text-to-video models.

### C.2  BASELINES

For different baselines, we follow their official implementations. **CFG++** does not require the guidance scale $w$ in the CFG. Instead, it employs a hyperparameter $(0.0 - 1.0)$ to implement guidance. To align other methods using standard CFG guidance, we map the $0 - 20$ guidance scale to $0.0 - 1.0$, which is the parameter required for CFG++. Moreover, to fit flow-based methods, we follow the authors' instructions in their official implementations [1]. For **APG**, we use the detailed implementation in their paper, with hyperparameters employed for DiT-XL/2, namely, $\eta = 0, r = 5, \beta = -0.5$. Please note that both CFG++ and APG are not designed for flow-based methods. For **CFG-Zero**, we directly adopt its official implementation [2] and use the default settings in SD3 pipeline.

---

[1]https://github.com/CFGpp-diffusion/CFGpp/issues/12
[2]https://github.com/WeichenFan/CFG-Zero-star

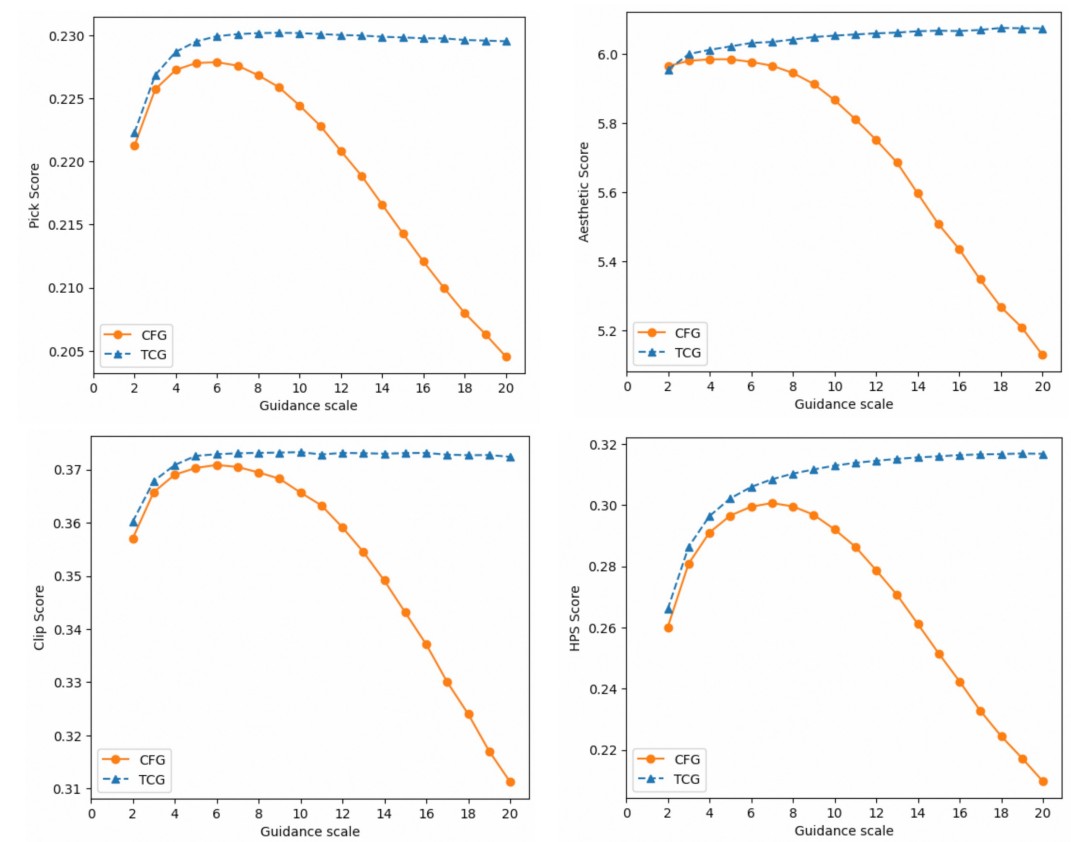

Figure 7: Results on the guidance scale from 2 to 20.

## D  ROBUSTNESS OVER WIDE GUIDANCE RANGE

Considering that TCG works well on high guidance scales, we investigate its robustness over a wide guidance range. As shown in Figure 7, the performance of CFG rapidly decreases at high guidance scales, while TCG works well across different guidance scales, demonstrating its robustness.

## E  VISUALIZATION ON MORE T2I BASE MODELS

In the main text, we provide visualization results for SD3.5 medium base model. Here we provide results of other base models. For **Flux-dev** base model, we provide qualitative results in Figure 8. For **SD3 medium** base model, we provide qualitative results in Figure 9. For **Lumina-Next** base model, we provide qualitative results in Figure 10.

For the results in Figure 1 of **SD3.5 medium** base model, the prompts from top to bottom are:

*There is a white toilet and a sink in this bathroom.*

*A brown cat crouches and arches its back in a white sink.*

*A vase with a flower growing very well.*

*Professional digital art of Godzilla with stunning detail.*

*Two colorful parrots perched together eating an egg tart.*

*A miniature anthropomorphic cat knight wearing pale blue armor and a crown.*

*A flat ink sketch of a hedgehog in the comic book style of Jim Lee.*

*Steve Buscemi portrays the Joker.*

## F LLM USAGE

We used a large language model (LLM) solely for language editing (e.g., grammar checks and readability improvements). It did not contribute to ideation, methodology, experimental design, or data analysis. All scientific content was developed by the authors. The authors take full responsibility for the manuscript and ensured that any LLM-edited text adheres to ethical guidelines and avoids plagiarism or scientific misconduct.

The image depicts a muscular mouse wielding assault rifles,
in a Disney art style.

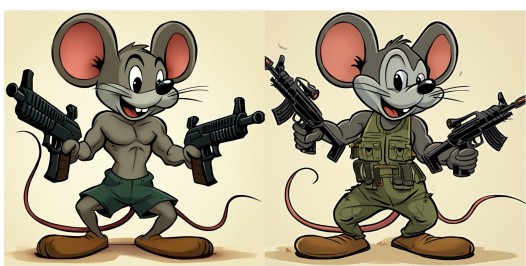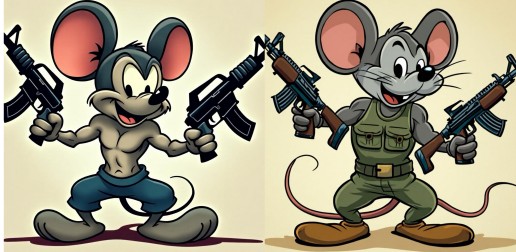

A dog resembling Hugh Laurie.

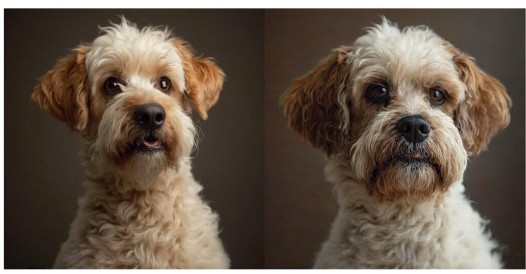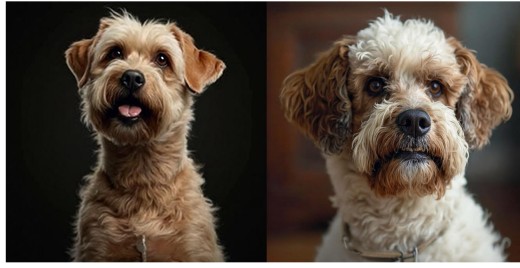

A portrait of a man resembling Super Mario against
a stylized background.

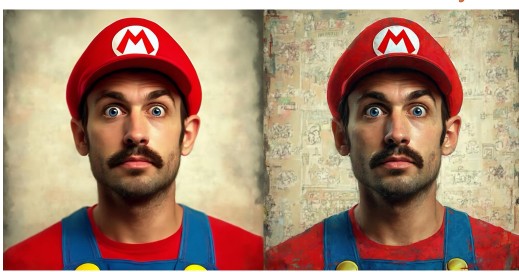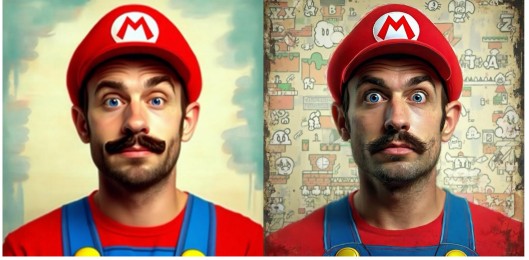

A digital painting of a favela city shrouded in mystical colors with radia
nt god rays and vibrant hues in the style of multiple artists.

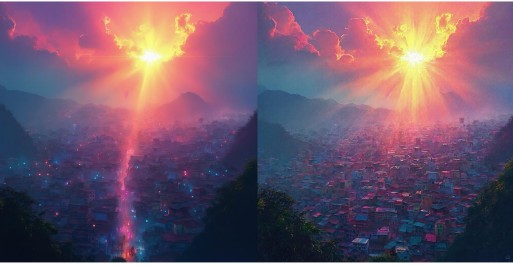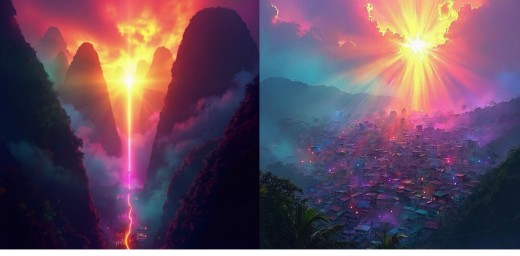

Animals fashioned from gems, colorful and shapely, depicted in natural
lighting, with a slight effervescence, artist credited to Alex Ross.

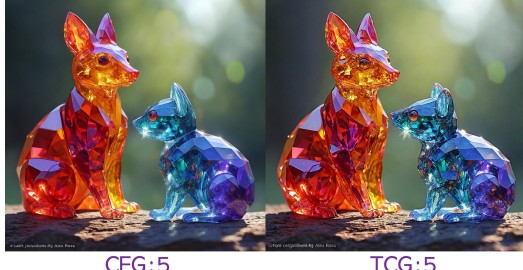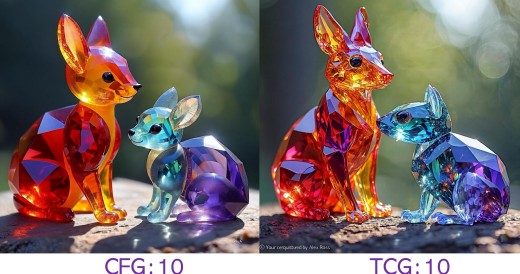

CFG:5                    TCG:5                    CFG:10                   TCG:10

Figure 8: Qualitative results on Flux-dev base model.

Anime art featuring Hatsune Miku with symmetrical shoulders.

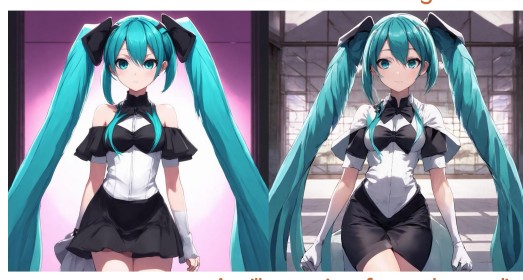 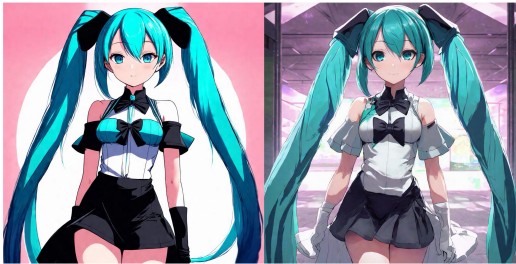

An illustration from the realistic comic book "Tiger White" featuring detailed artwork by a skilled illustrator.

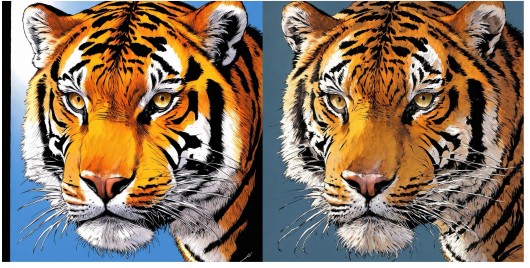 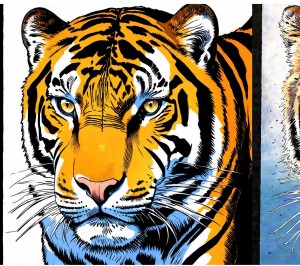 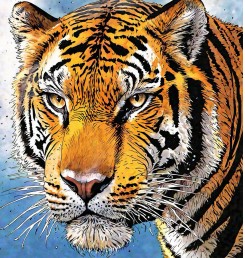

Doraemon is depicted as the Terminator using the Unreal Engine.

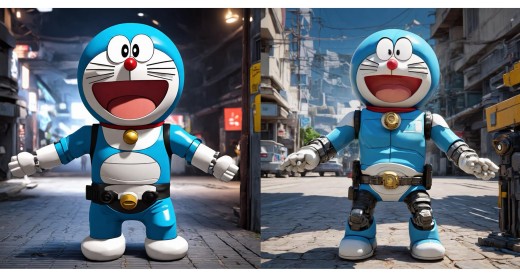 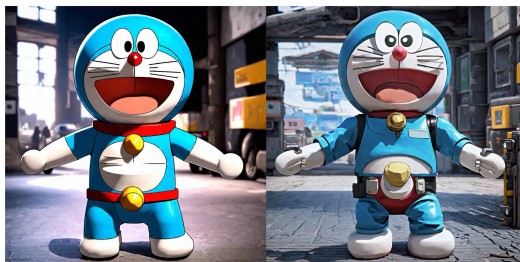

Darth Vader playing electric guitar on top of mountain.

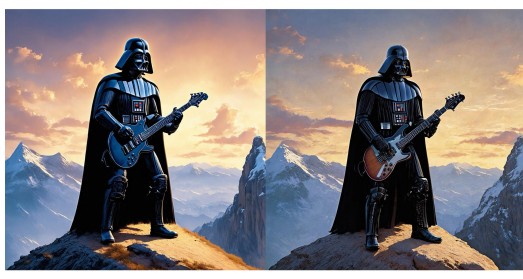 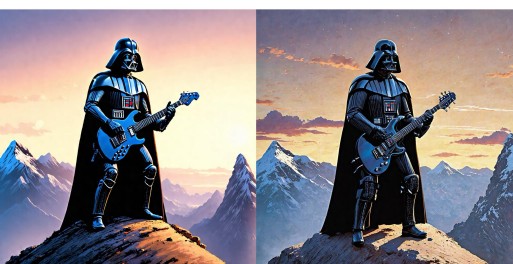

a man sitting on a motorcycle in the desert.

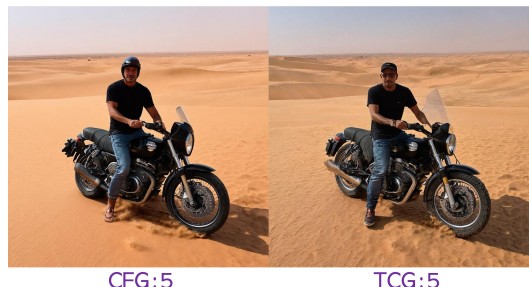 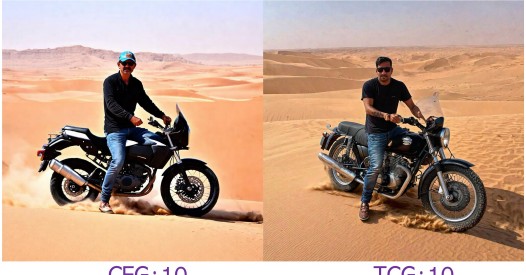

CFG:5       TCG:5       CFG:10      TCG:10

Figure 9: Qualitative results on SD3 medium base model.

A photo of Big Chungus from Looney Tunes.

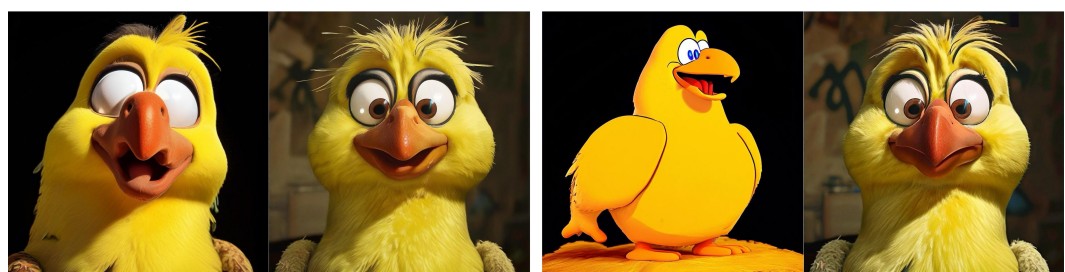

A man wearing a Batman costume holds a green glowing orb.

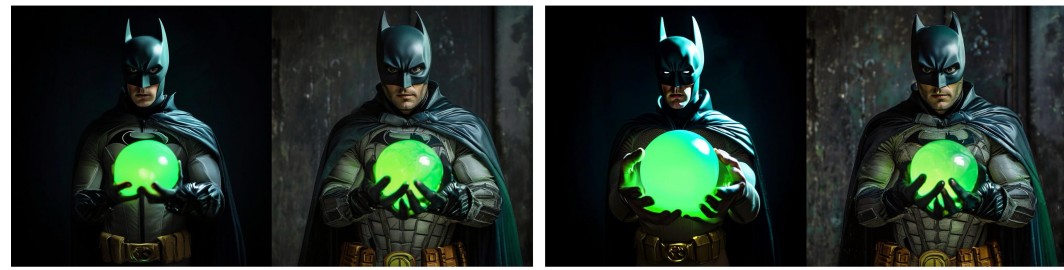

The image is a digital art headshot of an owlfolk character with
high detail and dramatic lighting.

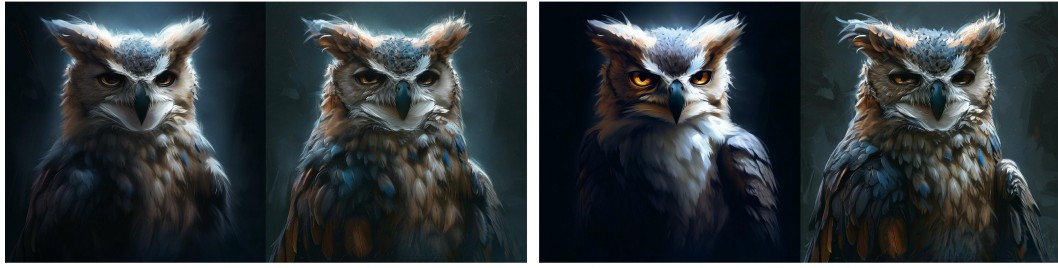

An anthropomorphic cat wearing sunglasses and a leather jacket
rides a Harley Davidson in Arizona.

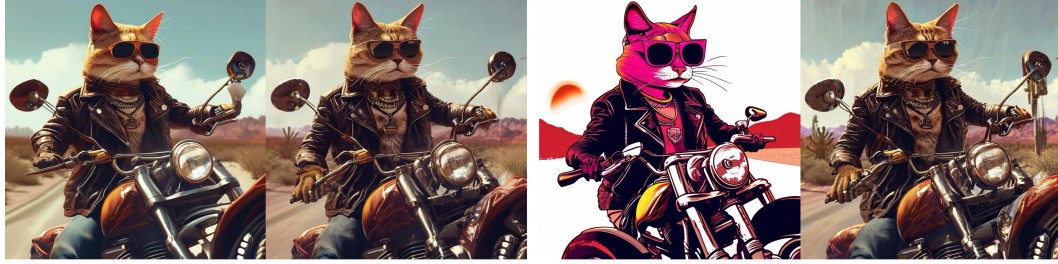

A little girl holding a brown stuffed animal.

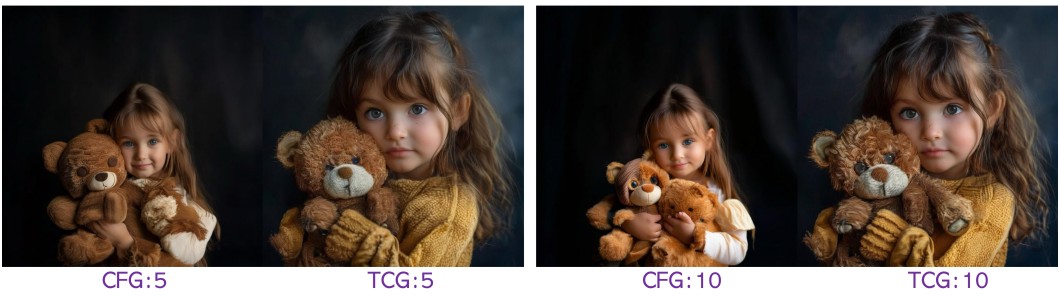

CFG:5            TCG:5            CFG:10            TCG:10

Figure 10: Qualitative results on Lumina-Next base model.

