# OpenReview forum: "TCG: Taming CFG for Flow Matching Models via Moment Matching and Adaptive Clipping"
_ICLR.cc/2026/Conference — ICLR 2026 Conference Withdrawn Submission_

### Official Review · Reviewer_oFoN · 2025-10-30

**Soundness:** 2
**Presentation:** 2
**Contribution:** 2
**Rating:** 4
**Confidence:** 4

**Summary:**

This paper investigates instability issues encountered in classifier-free guidance (CFG) for flow-based generative models, particularly at high guidance scales, which often lead to mode collapse and degraded visual results. The authors propose TCG, a plug-and-play inference-time guidance module comprising two key techniques: Moment Matching (MM), which aligns the first two moments of the velocity update to mitigate distribution shifts, and Adaptive Clipping (AdapC), which enforces temporal and spatial constraints on the guidance update to avoid spurious artifacts. Experiments on several text-to-image and text-to-video benchmarks, across multiple state-of-the-art models, indicate TCG outperforms standard CFG and recent alternatives in both quantitative metrics and qualitative quality.

**Strengths:**

- The paper presents a clear empirical diagnosis of why high-scale CFG causes generation artifacts in flow-based models, linking it to moment shifts in the velocity field and providing supporting visualizations (see Figure 2).
- The proposed solutions, MM and AdapC, are inference-time, training-free, light-weight, and broadly compatible with existing architectures, making them easy to adopt.
- Extensive experiments across several competitive benchmarks (HPD v2, DPG, GenEval, VBench) and a range of image/video generation models strengthen the empirical case for both improved robustness and upper-bound performance. Notably, Tables 1, 4, 5, 6, and 8 show consistent improvements in CLIP, Aesthetic, and prompt-alignment metrics for TCG across guidance scales.

**Weaknesses:**

- **Theoretical Depth & Lack of Formal Proofs:**

   While the paper provides intuitive and empirical arguments for moment matching and clipping, there is little in the way of rigorous formal analysis or proofs. Specifically, Section 3.1 alludes to moment mismatches causing out-of-distribution velocity predictions but lacks a theoretical derivation quantifying the connection between moment shifts and generation failure. The approach is therefore somewhat ad hoc, and the absence of formal convergence or generalization guarantees is a limitation for a work claiming to address fundamental instability. Many claims (especially those regarding the effectiveness of moment matching and clipping at stabilizing the guidance signal) are not explained by mathematical theory or backed by analysis of the optimization dynamics.

- **Ablation Depth on MM & Clipping and Hyperparameter Sensitivity:**

   While there are some ablations (e.g., Table 11, 12, Appendix B), the discussion around clipping threshold $\gamma$ and timestep $T_\text{clip}$ is relatively light. There is little analysis about failure cases, sensitivity to these hyperparameters, or potential adverse effects (e.g., when AdapC may degrade, oversmooth, or undercut the CFG signal).

- **Limited generation quality improvement**

**Questions:**

- Can the authors provide a more rigorous mathematical justification (even empirically motivated) for how moment-matching of the velocity field ties to the stability in generation, and why second-moment alignment is sufficient? Is higher-order moment alignment (e.g., skew/kurtosis) irrelevant or ineffective?

- Does Adaptive Clipping risk consistently suppressing important, high-magnitude (but semantically meaningful) variations, especially for complex or highly-structured prompts? Could this introduce bias or over-smooth outputs?

- How sensitive is the overall guidance quality to the choice of the $\gamma$ and $T_\text{clip}$ hyperparameters, especially in edge/corner cases?

**Details Of Ethics Concerns:**

N/A. The paper focuses on training-free modifications for generative model inference and does not raise ethical, societal, privacy, compliance, or unprofessional conduct concerns based on the content provided.

---

### Official Review · Reviewer_b1xN · 2025-10-31

**Soundness:** 2
**Presentation:** 3
**Contribution:** 2
**Rating:** 2
**Confidence:** 4

**Summary:**

This manuscript introduces TCG, a variant of classifier-free guidance for flow-based models. TCG adopts two key modifications of CFG:
- moment matching, which recalibrates the statistics of the guided velocity to align with the unconditional velocity,
- adaptive clipping, which clips the magnitudes of the guidance term using a temporal scheme and a spatial scheme.
The proposed method is tested and evaluated on various text-to-image models and a text-to-video model.

**Strengths:**

- Good intuition for the temporal clipping mechanism. The design choice to prevent spikes in guidance and enforce monotonically decreasing guidance term aligns with flow models' ideal behavior and is shown to be effective in the ablation studies.
- Quantitative results appear strong on the presented metrics. In particular, no significant degradation is observed from these metrics under a high guidance scale.

**Weaknesses:**

- The moment matching method aims to align the statistics of the guided velocity with the statistics of the unconditional velocity. This does make sense to me. The reason why CFG is required is because diffusion/flow models underfit the real velocity, thus both conditional/unconditional velocities tend to have smaller norms than the ground truths. This effect can be seen when measuring the saturation (used as a metric in APG) of the generated data w/o guidance, which would be usually much lower than the saturation of the real data. In other words, unguided generation tends to be undersaturated. Therefore, aligning with such statistics may result in deviation from real data. I would strongly suggest comparing the pixel statistics and saturation values of the generated data with real data on an imagenet model (e.g. SiT), like Fig. 7 in the APG paper.

- Qualitative results show high-frequency texture artifacts. In Fig. 9, TCG results contain excessive noisy texture when zoomed in. This also affects some fine-grained structures, like the hands and fingers in the Doraemon example in Fig. 9.

- Quantitative evaluation metrics are not strong enough to reflect the quality of fine-grained details. Most image metrics (except GenEval which focuses on objects not aesthetics quality) used in the paper are based on CLIP-like low-res ViT models, which can hardly judge the fine-grained details of the generated images. Thus, despite the strong metrics in the evaluation results, the texture artifact issue is not reflected in the quantitative evaluation. Adding metrics like patch FID could better measure the texture quality.

- The presentation of moment matching is unclear. It is never stated what underlying distribution the moments are computed from. Are the moments computed from spatial statistics (e.g., mean is the 2D spatial average of the velocity map for images)? Algorithm 1 also does not further clarify this.

**Questions:**

Please see weaknesses for rebuttal

---

### Official Review · Reviewer_ygzp · 2025-11-01

**Soundness:** 3
**Presentation:** 3
**Contribution:** 2
**Rating:** 6
**Confidence:** 4

**Summary:**

This paper studies why large classifier free guidance (CFG) scales destabilize flow matching models and proposes TCG, a plug and play, training free module with two parts: Moment Matching (MM), which stabilizes the velocity distribution by aligning its first two moments (mean and variance); and Adaptive Clipping (AdapC), which dynamically constrains the guidance update term from both temporal and spatial perspectives. Across SD3/SD3.5, Lumina Next, Flux dev, and Wan2.2, TCG improves robustness and slightly boosts metrics at moderate and high guidance scales.

**Strengths:**

+ Clear problem focus, practical fix. The paper targets a real pain point: instability and artifacts at high CFG scales in modern flow models. The diagnosis (distribution shift in predicted velocity) is well motivated and visualized (Fig. 2).
+ Coherent, simple, plug and play method. The combination of moment alignment in velocity space with temporal + spatial clipping is easy to insert into existing pipelines and requires no retraining.
+ Broad, careful evaluation. Experiments cover T2I and T2V, with multiple base models and standard benchmarks (HPD v2, GenEval, DPG, VBench). The paper includes ablations for MM vs. AdapC and for the two clipping strategies.
Weaknesses

**Weaknesses:**

- High CFG utility remains limited. Even when stabilized, increasing w beyond the usual sweet spot rarely yields clear gains; vanilla CFG performance often drops from $w=5 \to 10 \to 15$, and TCG largely mitigates the drop rather than unlocking a strictly better optimum (e.g., Table 1 and related tables). This narrows the practical significance: TCG keeps high w from “breaking,” but high w still doesn’t become consistently better than moderate w.
- Conceptual overlap with known normalization ideas. Moment alignment (zero centering + variance matching) mirrors long standing feature distribution alignment techniques. The contribution is a well engineered instantiation in velocity space, not a fundamentally new principle. That said, the specific formulation and coupling with dual clipping constitute a concrete, useful solution.
- Limited analysis of estimator noise. MM relies on online mean/variance per step. The paper does not quantify variance or bias of these estimates in single sample inference, nor compare to smoothed statistics (EMA) or reference/teacher statistics.
- Theory is light. The paper frames instability as a moment mismatch but stops short of a principled analysis of when/why moment control guarantees stability or preserves on manifold dynamics. The claims are supported empirically, not theoretically.

**Questions:**

- Generality beyond flows. A small study on SDE based samplers (DDPM/EDM) would increase generality, given the method’s training free nature.

---

### Note · Authors · 2025-11-13

**Comment:**

We appreciate the reviewers’ time and comments.

**Withdrawal Confirmation:**

I have read and agree with the venue's withdrawal policy on behalf of myself and my co-authors.